# Levetiracetam Reduced the Basal Excitability of the Dentate Gyrus without Restoring Impaired Synaptic Plasticity in Rats with Temporal Lobe Epilepsy

**DOI:** 10.3390/brainsci10090634

**Published:** 2020-09-11

**Authors:** Guillermo González-H, Itzel Jatziri Contreras-García, Karla Sánchez-Huerta, Claudio M. T. Queiroz, Luis Ricardo Gallardo Gudiño, Julieta G. Mendoza-Torreblanca, Sergio R. Zamudio

**Affiliations:** 1Laboratorio de Neurociencias, Subdirección de Medicina Experimental, Instituto Nacional de Pediatría, Ciudad de Mexico 04530, Mexico; gohegui@gmail.com (G.G.-H.); jatziri1984@hotmail.com (I.J.C.-G.); karlasanchezhuerta@hotmail.com (K.S.-H.); 2Departamento de Fisiología, Instituto Politécnico Nacional, Escuela Nacional de Ciencias Biológicas, Ciudad de Mexico 04530, Mexico; 3Area of Neurosciences, Department Biology of Reproduction Unidad Iztapalapa, Universidad Autónoma Metropolitana, Ciudad de Mexico 09340, Mexico; 4Brain Institute, Federal University of Rio Grande do Norte, Natal 59056-450, Brazil; clausqueiroz@neuro.ufrn.br; 5Servicio de Electromedicina, Instituto Nacional de Pediatría, Ciudad de Mexico 04530, Mexico; lrgallardo67@yahoo.com

**Keywords:** inhibitory transmission, synaptic plasticity, temporal lobe epilepsy, evoked field potentials, levetiracetam

## Abstract

Temporal lobe epilepsy (TLE), the most common type of focal epilepsy, affects learning and memory; these effects are thought to emerge from changes in synaptic plasticity. Levetiracetam (LEV) is a widely used antiepileptic drug that is also associated with the reversal of cognitive dysfunction. The long-lasting effect of LEV treatment and its participation in synaptic plasticity have not been explored in early chronic epilepsy. Therefore, through the measurement of evoked field potentials, this study aimed to comprehensively identify the alterations in the excitability and the short-term (depression/facilitation) and long-term synaptic plasticity (long-term potentiation, LTP) of the dentate gyrus of the hippocampus in a lithium–pilocarpine rat model of TLE, as well as their possible restoration by LEV (1 week; 300 mg/kg/day). TLE increased the population spike (PS) amplitude (input/output curve); interestingly, LEV treatment partially reduced this hyperexcitability. Furthermore, TLE augmented synaptic depression, suppressed paired-pulse facilitation, and reduced PS-LTP; however, LEV did not alleviate such alterations. Conversely, the excitatory postsynaptic potential (EPSP)-LTP of TLE rats was comparable to that of control rats and was decreased by LEV. LEV caused a long-lasting attenuation of basal hyperexcitability but did not restore impaired synaptic plasticity in the early chronic phase of TLE.

## 1. Introduction

Epilepsy is one of the most common and widespread neurological disorders, affecting more than 65 million people around the world [1]. Epilepsy is characterized by spontaneous and recurrent seizures that reflect temporary disruption of brain function due to excessive abnormal discharge of cortical neurons [2]. Temporal lobe epilepsy (TLE), in which epileptogenic activity originates from structures in the temporal lobe, is the most common form of focal epilepsy [3,4] and accounts for approximately 60% of all epilepsy cases in adults [5]. TLE is the most common drug-resistant focal epilepsy disorder [6,7], and seizures are not controlled by medical treatment in up to 30% of these patients [8]. Therefore, studies are needed to investigate new drug targets for the treatment of epilepsy and its comorbidities.

Another concerning aspect of TLE is its cognitive sequelae due to specific deficits in temporal lobe-related functions [9]. Clinical and animal studies show that TLE can cause neurodegeneration of the hippocampus associated with cognitive and behavioral disturbances [10,11], affecting many aspects of cognitive functioning, including learning, memory, language, attention, executive functions and problem solving [9,12,13]. The underlying mechanisms of such impairments are not yet well understood.

Learning and memory are thought to emerge from the elementary properties of chemical synapses, such as synaptic plasticity. Paired-pulse facilitation and depression, as well as long-term potentiation (LTP), are forms of synaptic plasticity, representing some of the cellular phenomena that are used in neuronal networks for the processing of new information and its consequent translation to various behaviors [14,15,16]. Diverse alterations in short-term synaptic plasticity (facilitation/depression profile) and LTP have been reported in the CA1 and CA3 hippocampal regions of rats with TLE induced with pilocarpine [17,18,19]. Nevertheless, little information is known about TLE and the impairment in the synaptic plasticity of the dentate gyrus (DG), a region that is relevant due to the multiple changes in its synaptic connectivity during epilepsy [2].

Levetiracetam (LEV; S-enantiomer pyrrolidine derivative of α-ethyl-2-oxo-1-pyrrolidine acetamide, IUPAC, (S)-2-(2-oxopyrrolidin-1-yl)butanamide) is a second-generation antiepileptic drug that is widely used in patients with either generalized or focal seizures [20]. Although the exact mechanisms through which LEV controls seizures remain unclear, its anticonvulsive properties are thought to rely on its binding with synaptic vesicle protein 2A (SV2A) [21,22]. SV2A is an integral membrane protein present on the synaptic vesicles of nerve terminals, independent of their neurotransmitter content [23,24]; it regulates action potential-dependent neurotransmitter release by modulating exocytosis and endocytosis processes [25,26,27].

It is believed that LEV principally antagonizes the effect of SV2A, inhibiting it from participating in its usual role of priming vesicles and, as a result, generating a decrease in excitatory and inhibitory synaptic transmission, ultimately affecting neural excitability [28,29,30]. Indeed, post-status epilepticus (SE) acute or chronically LEV-treated rats showed retardation or decreases in various altered electrophysiological parameters in different hippocampal regions, such as neuronal synchronization and burst firing, the interictal spike rate or the occurrence of high-frequency oscillations ripples and fast ripples [20,31,32,33]. Furthermore, LEV has been implicated in the reversal of cognitive and synaptic dysfunctions, e.g., chronic treatment with LEV restored deficits in learning and memory and decreased LTP in the DG and the CA1 in an Alzheimer’s disease model with frequent abnormal spiking activity and intermittent seizures [34]. Likewise, LEV improved the impairment of spatial memory and reversed the decrease in LTP in the CA1 region in post-SE rats [18]. However, the long-lasting effect of LEV treatment and its participation in the synaptic plasticity of the DG, once epilepsy is debuted, have not been explored. Thus, the objectives of this study were to characterize the alterations in synaptic plasticity in the DG of rats with early chronic epilepsy and to determine whether LEV treatment restores the changes in basal excitability, short-term (facilitation/depression) and long-term (LTP) synaptic plasticity in this hippocampal region.

## 2. Materials and Methods

Twenty-nine male Wistar rats (250–300 g; Envigo, México) were used. Animals were housed under the following standard conditions: regulated temperature (22 ± 2 °C), light/dark cycle (12:12 h, lights on at 06:00) and food and water available ad libitum. Animals were randomly allocated to the following four groups: control (CONT, *n* = 7), epileptic (EPI, *n* = 9), CONT + LEV (*n* = 6) and EPI + LEV (*n* = 7) groups. All procedures followed the National Institutes of Health Guide for the Care and Use of Experimental Animals and the Mexican official Norm of the Agriculture Secretary (SAGARPA NOM-062-Z00-1999) published in 2001. The protocol was approved by Comité Institucional para el Cuidado y Uso de los Animales de Laboratorio (CICUAL) for animal experimentation (CICUAL INP-064/2015). The following experimental procedures were employed (Figure 1): SE (see Section 2.1) was induced in the EPI groups (EPI and EPI + LEV groups), while nonepileptic controls (CONT and CONT + LEV groups) were kept under similar conditions as the EPI rats, except that SE was not induced. Three weeks after SE induction, animals from the EPI and EPI + LEV groups were video monitored to record the first spontaneous behavioral seizure (see Section 2.2). After one seizure was detected in the video monitoring, EPI + LEV and CONT + LEV animals were implanted subcutaneously with an osmotic minipump (see Section 2.3). The minipumps delivered LEV (300 mg/kg/day, 10 µL/h) for 7 days. After that period, the minipumps were surgically removed. Electrophysiological recordings were conducted during the washout period, i.e., four days after the end of LEV treatment (see Section 2.4). The electrophysiological measurement procedure began with the positioning of the stimulation and recording electrodes, followed by a period of stabilization (120 min). After that, input/output (I/O) curves, paired-pulse (PP) and LTP induction protocols were implemented for electrophysiological measurements

### 2.1. Pilocarpine-Induced Status Epilepticus

SE was induced by systemic injection of pilocarpine, as previously described [35,36,37]. Briefly, animals were pretreated with lithium chloride (127 mg/kg, i.p.; Sigma-Aldrich, Ciudad de México, México) 22 h before pilocarpine administration. On the day of SE induction, animals were injected with scopolamine methyl-bromide (1 mg/kg, i.p.; Sigma-Aldrich, México) to avoid peripheral cholinomimetic effects, and 30 min later, they received a single dose of pilocarpine hydrochloride (30 mg/kg, i.p.; Sigma-Aldrich, México). Behavioral seizures were scored according to the Racine scale [38]; SE was defined as sustained convulsive behavior (stage 4 or 5 on the Racine scale) for more than 30 min [20]. Ninety minutes after SE began, rats were administered an intramuscular (i.m.) injection of 5 mg/kg of diazepam (PISA, Ciudad de México, México) and were placed on an ice bed for 1 h to reduce the hyperthermia produced by SE. CONT rats received saline solution (NaCl 0.9%) instead of pilocarpine. A second dose of diazepam was administered eight hours later, and then the rats received a rehydrating injection of saline solution (5 mL, 0.9%, subcutaneous (s.c.)) on the first night. Finally, the rats were housed overnight in a room at 17 ± 0.5 °C. Beginning one day after SE, the room temperature was restored to 22 ± 2 °C.

### 2.2. Monitoring of Spontaneous Behavioral Seizures

A previous study in our laboratory demonstrated that spontaneous behavioral seizures are established approximately 18 days after SE induction [37]. Therefore, video monitoring of seizures began 3 weeks after SE induction. Animals were housed in individual polycarbonate cages and were video monitored to record the occurrence of spontaneous seizures. Video monitoring was performed with four cameras (Steren Model CCTV-970, Mexico City, México), and the recordings were collected during the light period (08:00 to 17:00 h) [36,39]. The videos were analyzed by trained observers using the fast-forward (6×) component of the system. When seizure-like activity was detected, the video was reversed to the start of the behavior and examined at real-time speed. An animal was considered to have a seizure when the Racine score reached 4 or 5 points [37,38].

### 2.3. Levetiracetam Treatment

Two days after the first behavioral spontaneous seizure was observed, an ALZET^®^ osmotic minipump was implanted subcutaneously for one week to provide subchronic treatment with LEV (300 mg/kg/day). The LEV dose was chosen based on previous experiments in EPI rats [32,36]. LEV treatment via this route has been shown to lead to adequate LEV concentration in blood and proper LEV washout after removing the osmotic minipump [36]. To fill the osmotic minipump chambers, LEV was extracted from tablets (Keppra^®^). Briefly, two tablets of LEV were dissolved in 3 mL of saline solution (0.9%). Then, the mixture was sonicated and centrifuged for 15 min at 3000 rpm (1400× *g* centrifuge, Hermle Labnet 2326 K, rotor 220.72), the supernatant was filtered with a Corning 28 mm syringe filter of 0.45 µm, and finally, the osmotic minipumps were filled according to the manufacturer’s instructions.

### 2.4. Electrophysiological Recordings

On the recording day, four days after the removal of the osmotic minipumps, rats were anesthetized with urethane (1.3 g/kg; i.p.) and, after complete loss of reflexes, placed on a stereotaxic apparatus. A heating pad was used to maintain body temperature at 37 ± 0.5 °C. To register local field potentials, a stainless steel concentric electrode with a tip diameter of 250 μm was placed in the dorsal DG (anteroposterior (AP) −3.5 mm, mediolateral (ML) 2.0 mm and dorsoventral (DV) −3.0 to −3.4 mm from the dura), and another stainless steel concentric bipolar stimulation electrode was placed in the perforant path (AP −7.2 mm, ML 4.1 mm and DV −2.4 to −3.2 mm from the dura) [40] (Figure 2). Final adjustments in the DV coordinates of both electrodes were made to produce an evoked potential of optimal morphology; then, a stabilization protocol with a duration of 2 h was performed: a series of 5 single square pulses (0.1 ms in duration) and 1500 µA of intensity were delivered to the perforant path at 15 min intervals using a Grass S-88 stimulator and a PSIU-6 constant current isolator (Grass Technologies, West Warwick, RI, USA). Evoked local field potentials were amplified (gain: 200), digitized (10 kHz sampling rate) and stored in a Biopac MCE 100C system (Biopac Systems Inc., Goleta, CA, USA). Briefly, evoked local field potentials recorded in the DG were composed of an initial positive element that corresponded mainly to the excitatory postsynaptic potential (EPSP) and a negative component that represented the sharp population spike (PS) (Figure 3, top right). The EPSP magnitude is associated with synaptic dendritic activity, and the PS magnitude represents the number of granule cells producing action potentials as a result of the EPSPs (Figure 3, top right). The magnitude of the EPSP was measured as the slope of the rising phase of the potential prior to PS onset. PS amplitude was calculated by the magnitude of a line connecting the lowest value of the negative component with a line connecting the two positive peaks (Figure 3, top right) [41,42].

#### 2.4.1. Input/Output Curves

I/O curves were constructed using stimulation intensities ranging from 50 to 1500 µA (monophasic pulse duration: 0.1 ms). Averaged evoked potentials (*n* = 5) for each intensity were used for quantification. Only animals in which the maximum applied intensity evoked EPSPs of 3 mV or higher were used for further analysis. Both EPSP and PS I/O curves reflect the excitability of the circuit under basal conditions by evaluating the relationship between the intensity of the stimulus supplied to the perforant path and the magnitude of the electrophysiological responses in the DG. Additionally, the latencies of EPSP and PS, at intensities that produced 50% or 100% of maximal PS, were taken from the stimulation artifact to EPSP onset, PS onset, PS peak or EPSP peak (Figure 5 right). We define EPSP onset as the point at which the amplitude increases from the baseline recording; meanwhile, PS onset was defined as the point at which the amplitude decreases from the ascendant component of the EPSP [41,43].

#### 2.4.2. Paired-Pulse Depression and Facilitation

Field potentials were evoked by PP stimulation in the perforant path and recorded in the DG at intensities that produced 20, 50 and 100% of maximal PS from the I/O curve, with interpulse intervals (IPIs) of 10, 20, 30, 70 and 250 ms. PP stimulation examines both facilitation (PPF) and depression (PPD) [41]. Data are represented as a PP percentage ((pulse 2 amplitude/pulse 1 amplitude) × 100), a percentage < 100 reflects depression and a percentage > 100 reflects facilitation.

#### 2.4.3. Long-Term Potentiation

To induce LTP, high-frequency stimulation was delivered into the perforant path, and the response was examined in the DG. Tetanic stimulation consisted of three train pairs at 400 Hz, four stimuli per train, separated by 200 ms between each four-pulse burst, and 10 s between each train pair at a maximal stimulation of 1500 µA. Pre- and post-train stimuli of an intensity that produced 50% of maximal PS were delivered to evaluate LTP, responses were recorded 15 min before and 5 min after tetanic stimulation delivery, and LTP measures continued every 15 min for 125 min (modified from Sánchez-Huerta et al. [41]). The EPSP slope and the PS magnitude are expressed as percentages of baseline (pretrain) values.

### 2.5. Histological Verification

At the end of the electrophysiological experiments, animals were transcardially perfused with saline solution (0.9%) followed by buffered paraformaldehyde (4%). Then, the rats were decapitated, and their brains were dissected and postfixed at room temperature for 12 h in the same fixative medium. Next, the brains were transferred to sucrose (30%; diluted in phosphate buffer) until complete infiltration. Finally, serial coronal sections (150 μm thick) were cut with a vibratome (Electron Microscopy Sciences USA, model OTS-4000). The slices were mounted on gelatinized slides and stained with cresyl violet (Sigma-Aldrich). Cytoseal (Electron Microscopy Sciences, Hatfield, PA, USA) was added, and the brain sections were observed using a clear field microscope (Olympus BX51) equipped with a digital video camera (mbf CX9000) to verify the location of the electrodes. Data were collected and processed only from those animals in which the stimulation and recording electrodes were correctly placed.

### 2.6. Statistical Analysis

Latencies were analyzed using a two-way analysis of variance (ANOVA) with two between-subject factors: EPI condition and treatment. Linear regression analysis of PS amplitude as a function of the EPSP slope curve was performed, and Student’s *t*-test was used to compare slopes. Statistical analyses were performed with SigmaStat 3.5 software (SigmaStat 3.5 software, Systat Software, Inc. San Jose, CA, USA). Data from the I/O curve, PPF, PPD and LTP were analyzed by three-way repeated-measures (RM) ANOVAs (an ad hoc Excel worksheet was used), with one within-subject factor (intensity, IPI or time, respectively) and two between-subject factors (EPI condition and treatment). When appropriate, a Student–Newman–Keuls (S–N–K) test was used as a post-hoc comparison test. Finally, *p*-values of 5% or lower were considered to be statistically significant. All data are expressed as the mean ± standard error of the mean (S.E.M.).

## 3. Results

### 3.1. Basal Excitatory Synaptic Transmission: Input/Output Curve

The evoked local field potentials were recorded in the DG of the hippocampus (Figure 2), and they were measured as the EPSP slope and the PS amplitude (Figure 3, top). Analysis of the I/O curve over a range of stimuli represents the state of synaptic basal excitatory transmission [41]. At all intensities tested, the magnitude of the EPSP slope, associated with synaptic dendritic activity, was not affected by TLE or treatment (Figure 3, top), and three-way RM ANOVA did not reveal significant differences in the main effects of treatment or EPI condition. However, as a consequence of the stimulus–response relation, a significant difference in the within-subject factor (intensity) was found (F_15,330_ = 52.63, *p* < 0.01). Nevertheless, the statistical analysis did not reveal a significant interaction among treatment, EPI condition and intensity factors.

In contrast, the three-way RM ANOVA for the I/O curve of PS amplitude revealed significant differences in the EPI condition (F_1,22_ = 24.63, *p* < 0.01), treatment (F_1,22_ = 6.86, *p* < 0.05) and intensity (F_15,330_ = 39.19, *p* < 0.01) factors, without an interaction among these three factors. Post-hoc S–N–K tests revealed that the EPI and EPI + LEV rats showed increased PS amplitude compared with the CONT and CONT + LEV groups. This increase in PS magnitude was significant at stimulus intensities from 100 to 1500 µA for EPI rats and from 200 to 1500 µA for the EPI + LEV group (Figure 3, bottom). Interestingly, when the EPI and EPI + LEV groups were compared, a significantly higher PS magnitude at intensities from 800 to 1500 µA was observed in the EPI rats vs. the EPI + LEV group (Figure 3, bottom).

PS amplitude as a function of EPSP slope is represented in Figure 4; the excitability curves were similar for CONT and CONT + LEV rats. The linear regression analysis revealed significant correlations with positive slopes in both groups (CONT: b = 1.00, *R*^2^ = 0.87, *p* < 0.01; CONT + LEV: b = 0.96, *R*^2^ = 0.86, *p* < 0.01), without a significant difference in their slopes. EPI groups also showed significant correlations (EPI: b = 2.47, *R*^2^ = 0.97, *p* < 0.01; EPI + LEV: b = 1.91, *R*^2^ = 0.94, *p* < 0.01) without statistical significance in their slopes. Interestingly, the EPI condition corresponded to a shift to the left of the excitability curve compared with the control condition; higher slope values were found in EPI animals than in CONT animals (EPI vs. CONT, t_8_ = 5.06; *p* < 0.01; EPI + LEV vs. CONT, t_8_ = 2.34; *p* < 0.05). Although LEV treatment did not prevent this shift in the curve in EPI rats, the higher PS amplitudes observed in the EPI group were absent in EPI + LEV rats (Figure 4).

The latencies of EPSP onset and the EPSP peak were similar among all groups (Figure 5) at intensities that produced 50% or 100% of the maximal PS according to the I/O curve. Two-way ANOVA failed to detect any main effects of treatment on EPSP onset at 50% and 100% and for the EPSP peak at 50% and 100% of maximal PS from the I/O curve. There were also no significant main effects of the EPI condition factor or the treatment × condition interaction for EPSP onset or EPSP peak latencies at both intensities tested.

In contrast, two-way ANOVA revealed significant main effects of EPI condition on the latencies of PS onset (50%; F_1,25_ = 27.33, *p* < 0.01, 100%; F_1,25_ = 18.65, *p* < 0.01) and PS peak (50%; F_1,25_ = 15.13, *p* < 0.01, 100%; F_1,25_ = 11.61, *p* < 0.01) for the 50% and 100% maximal responses. The post hoc S–N–K tests detected significant differences, and the EPI and EPI + LEV groups showed lower latencies for PS onset and PS peak than both the CONT and CONT + LEV groups (Figure 5). There were no significant main effects of treatment or significant effects of the treatment × condition interaction for the PS onset and PS peak latencies at both intensities tested.

### 3.2. Short-Term Plasticity: Paired-Pulse Facilitation and Depression

Paired stimulation of the perforant path produces, depending on the IPI, facilitation or depression of the evoked field potentials recorded in the DG. These synaptic changes are referred to as short-term plasticity, where PPF reflects the pre- and postsynaptic modulation; in turn, PPD reveals the integrity of the local inhibitory circuits (GABAergic interneurons) [44]. Although PP stimuli were delivered at intensities that produced 20, 50 and 100% of the maximal PS according to the I/O curve, the profiles of facilitation and depression were better established at 100% of the maximal response; thus, this intensity was chosen for analysis.

Paired stimulation at different IPIs resulted in a facilitation/depression pattern in the CONT and CONT + LEV groups (Figure 6). The three-way RM ANOVA revealed significant differences in the EPI condition, the between-subject factor (F_1,25_ = 9.69, *p* < 0.01) and in IPI, the within-subject factor (F_4,100_ = 35.65, *p* < 0.01), but not in the treatment factor or in the IPI × treatment × EPI condition interaction. The S–N–K test revealed significant differences in CONT groups compared with the EPI groups at short (10 and 20 ms) and intermediate (30 and 70 ms) IPIs (Figure 6). In the CONT group, PPD was observed at short IPIs (10 and 20 ms) and in the longer interval (250 ms), and PPF was observed at intermediate intervals (30 and 70 ms), reaching the maximal facilitation with 70 ms IPI. This triphasic pattern (PPD–PPF–PPD) was not observed in EPI animals. EPI rats did not show PPF at intermediate IPIs; instead, PPD was observed at 30 ms intervals, and neither facilitation nor depression was observed at an IPI of 70 ms. Interestingly, the EPI group showed a higher PPD at short IPIs (10 and 20 ms) than the CONT groups. This altered pattern in short-term plasticity caused by epilepsy was not modified by LEV treatment, as EPI + LEV rats showed the same profile of short-term plasticity as EPI rats.

### 3.3. Long-Term Plasticity

LTP is a long-lasting increase in the strength of synaptic transmission [45]. Here, LTP was elicited in the DG by high-frequency stimulation of the perforant path. After delivering tetanic stimulation, a modest (~25%) increase in the EPSP slope in the CONT, CONT + LEV and EPI groups was apparent (Figure 7, top), which was of minor magnitude in the EPI + LEV group (main effects of treatment: (F_1,22_ = 36.97), *p* < 0.01 and S–N–K test). This mild potentiation of the evoked EPSP slope decayed over time and was statistically significant for all groups at 95, 110 and 125 min compared with at 5 min post-train (time factor: F_8,176_ = 5.24, *p* < 0.01 and S–N–K test). Three-way RM ANOVA did not reveal any significant differences for the main effects of the EPI condition or the EPI condition × treatment × time interaction.

At a difference in the LTP of the EPSP slope, the CONT and CONT + LEV groups had robust (~250%) LTP in the PS magnitude after train; both EPI groups also had PS-LTP, albeit to a minor degree (~180%; Figure 7, bottom). This difference between the CONT groups and the EPI groups was significant, and three-way RM ANOVA revealed significant main effects of EPI condition (F_1,21_ = 11.17, *p* < 0.01). The LTP of PS magnitude also declined over time in all groups (time factor: F_8,168_ = 7.86, *p* < 0.01), and the S–N–K test revealed significant differences from 50 to 125 min after tetanic stimulation compared with 5 min post-stimulation. However, LEV treatment was not able to prevent the alteration in PS-LTP caused by epilepsy, and no significant differences were detected by three-way RM ANOVA for the main effects of treatment or the epileptic condition × treatment × time interaction.

## 4. Discussion

In this study, we characterized the alterations in basal excitability, facilitation, depression and LTP in the DG of the hippocampus of rats with early chronic epilepsy and determined the long-lasting effect of LEV treatment. The main findings of this research were that, in the I/O curve, EPI animals presented an increase in the amplitude of PS and a reduction in the onset- and peak-PS latencies with respect to nonepileptic groups. Interestingly, LEV treatment partially reduced this increase but did not lower PS amplitude to CONT levels. In turn, TLE caused an augmentation in PPD without showing PPF. Nevertheless, LEV treatment in EPI rats did not alleviate such alterations. Finally, animals in the CONT, CONT + LEV and EPI groups showed mild EPSP-LTP with a decrease in the EPI + LEV-treated group. With respect to PS-LTP, nonepileptic groups showed a robust response that was reduced in EPI animals; LEV treatment was not able to restore this reduction in EPI rats. It is important to mention that all our results reflect the semipermanent changes caused by LEV in the neurochemistry of the system, since the recordings were realized four days after the cessation of LEV treatment. This long-lasting effect of LEV has been previously reported for LEV and some other antiepileptic drugs but not for all our electrophysiological parameters studied [32,36,46,47].

### 4.1. Hyperexcitability Caused by TLE in the DG Is Attenuated by LEV Treatment

The effects of LEV on basal synaptic transmission in the hippocampal DG area were examined before the PPF, PPD and LTP experiments. I/O curves represent basal synaptic transmission, reflecting not only the level of presynaptic neurotransmitter release but also postsynaptic processes [44]. Rises in stimulation intensity typically result in an increased EPSP slope and an elevated PS amplitude, as observed for all groups. However, TLE caused a significant augmentation of PS amplitude and a reduction in the onset- and peak-PS latencies, indicating clear signs of hyperexcitability in the DG in the early chronic phase of epilepsy. This finding is consistent with previous findings where an increase in PS amplitude in the DG area of post-SE anesthetized rats was observed [32] and with a reduction in the onset- and peak-PS latencies reported in freely moving kainate-induced TLE animals [33]. Interestingly, when PS amplitude was plotted as a function of EPSP slope, the EPI condition caused a shift to the left in the excitability curve compared with controls, showing that coupling between the EPSP and PS reflects the final result of the neuronal synaptic response [48]; our EPSP-PS data reinforce the idea of a hyperexcitable state in the granule cells of the EPI brain. The hyperexcitability of dentate granule cells could be a consequence of pathological rearrangements of neuronal circuitry on which an initial loss of hilar mossy cells denervates granule cell dendrites; this triggers the formation of abnormal recurrent excitatory connections among normally unconnected granule cells (mossy fiber sprouting, Figure 8a) [49]. Additionally, there is a combination of GABAergic hilar interneuron loss and connectivity alterations in the remaining interneurons (Figure 8a,b) [2,49,50]. This imbalance between excitation and neuronal inhibition may be the origin of the hyperexcitable and hypersynchronous neuronal activity observed.

Our data also demonstrated that LEV treatment partially reversed the hyperexcitability of the DG in the chronic phase of epilepsy; these results extend upon the findings of Margineanu et al. [32], who reported that LEV treatment inhibited the development of hippocampal DG hyperexcitability in the epileptogenic phase of pilocarpine-induced epileptic rats. The mechanism of action through which LEV is able to reduce DG excitability remains unknown; however, the mechanism may involve effects on excitatory and inhibitory neurotransmitter release [29,53] since the LEV primary target is SV2A protein, which is expressed in all nerve terminals independently of their neurotransmitter content and is involved in modulation of the vesicular cycle [21,24,30]. Nevertheless, data suggest that LEV has a selective effect on the DG inhibitory system, as SV2A protein is strongly coexpressed with GABAergic markers under healthy conditions [24,54] and preferentially regulates vesicular γ-aminobutyric acid (GABA) release in the hippocampus [55,56]. In addition, under EPI conditions, SV2A is increased and coexpressed with GABAergic, but not glutamatergic, markers in the hilar interneurons, suggesting that SV2A specifically regulates GABAergic neurotransmission in the hilus as a compensatory antiseizure mechanism [54,57]. Furthermore, this scenario is consistent with the results reported by Pichardo-Macías et al. [36], who showed that LEV treatment might re-establish the balance in the glutamate/GABA ratio, increasing the vesicular release of GABA in the chronic phase of TLE induced by lithium–pilocarpine treatment. Therefore, LEV may act as an effective antiseizure agent that potentiates inhibitory transmission, enhancing GABA release and suppressing the firing of glutamatergic neurons in the DG (Figure 8c).

### 4.2. TLE Caused Alterations in Short-Term (Facilitation/Depression) Synaptic Plasticity, and LEV Did Not Reverse Them

As previously mentioned, PPD reflects the integrity of GABAergic circuits, which are constituted by different populations of interneurons innervating the granule cells of the DG [58]. Our results showed that EPI rats exhibit increased PPD; this observation is in line with previous reports in kainate-induced SE rats [33,59] and supports the existence of an elevated GABAergic tone in the DG of EPI animals during the chronic phase of this disease. The mechanism associated with the hyperactivation of GABAergic networks remains intriguing. Although there is a consensus that chronic epilepsy is associated with a loss of different subtypes of GABAergic interneurons in the hilus of rodents and humans (Figure 8a,b) [51,52,60], in addition to the death of some interneurons, plastic changes in the inhibitory networks and altered postsynaptic responses of the remaining neurons could also influence GABAergic tone (Figure 8b). In this regard, it has been reported that calbindin-immunoreactive interneurons show enlargement of their cell bodies, growth of numerous spines and elongation of dendrites (Figure 8b) [52]. Additionally, during mossy fiber sprouting, some interneurons are targeted by axons of granule cells, establishing aberrant networks (Figure 8b) [49,51]. Moreover, patch-clamp recordings have shown postsynaptic alterations, evidencing that tonic GABA currents, mediated by lower affinity GABA_A_ receptors, are enhanced in granule cells of EPI rats [61]. These structural and functional changes could partially explain the increased GABAergic tone in the DG of EPI rats. Nevertheless, the functional significance of this alteration needs to be addressed; specifically, elevated inhibitory activity could be responsible for synchronizing granule cells, thus contributing to the generation of seizures, or this alteration could simply control the efficacy of excitatory inputs and thereby limit synaptic plasticity [62].

On the other hand, our results indicate an absence of PPF in rats with TLE. PPF reflects a short duration increase in synaptic transmission derived from pre- and postsynaptic modulation [44]. It has been reported that facilitation may depend on several factors, such as residual calcium, vesicular readily releasable pool increases, properties of postsynaptic receptors, and synaptic activity frequency [63]. Although not all these mechanisms have been studied in EPI rats, Upreti et al. [64] observed an increase in the number of vesicles of the readily releasable pool and more vesicular release and endocytosis in granule cells of EPI rats. These changes are consistent with an augmentation in glutamate release in EPI rats under basal and depolarizing conditions [36,65], which in turn could modify the expression of glutamate transporters, increasing the uptake of that neurotransmitter in the hippocampal region [66,67]. These data suggest persistent glutamatergic activity in the presynaptic compartment, which could explain the greater PS magnitude observed in this study; however, such exacerbated activity could exhaust neurotransmitter availability when PP is applied at short IPIs. Furthermore, glutamatergic activity could explain not only the loss of facilitation by PP but also the persistence of GABAergic tone in interneurons, which causes a hyperpolarized environment that prevents an increased response to the second stimulus. A third possibility is a combination of these processes.

The EPI + LEV group did not show any change with respect to the EPI group, as PPD reflects the influence of the inhibitory postsynaptic potentials that are produced by the GABA released from synaptic vesicles of interneurons, the subsequent activation of postsynaptic GABA_A_ receptors and the entrance of Cl^-^ ions, which increases the negative charge inside the postsynaptic neuron [2,44]. The resultant augmentation in membrane conductance and hyperpolarization underlies what is known as phasic inhibition [68,69]. GABA can also activate other receptors on presynaptic terminals or at neighboring synapses, causing persistent tonic activation of GABA_A_ receptors [69,70]. This kind of activation may affect the magnitude and duration of the response to a stimulus, reducing the probability that an action potential might be generated [69]. Since PPD and PPF are typically measured with PS amplitude, the continuous activity of interneurons could provoke the absence of PS in the PP procedure and explain the strong augmentation of PPD and the absence of PPF in EPI rats. The lack of effect of treatment with LEV to restore PPD and PPF in EPI rats could be due to a “floor effect”, EPI rats exhibit an elevated GABAergic tone in the DG, which seems to have reached its maximum level, thus avoiding the LEV effect.

### 4.3. TLE Reduced PS-LTP, and LEV Did Not Correct This Alteration

Likewise, our results showed that in the DG of the hippocampus of early chronic EPI rats, the EPSP-LTP slopes were comparable to nonepileptic animals; however, there was a decrease in PS-LTP amplitude. Early LTP is a process that depends on glutamate release, the repeated activation of the AMPA receptors and subsequent activation of NMDA channels that allow Ca^2+^ influx and the activation of some kinase enzymes (e.g., PKC, CAMKII and Fyn) [44,71]. In chronic epilepsy, the vesicular release of glutamate is substantially increased in the DG, resulting in elevated extracellular levels of this neurotransmitter after high-K^+^ stimulation [36,65,72]. Nevertheless, the NR2B subunit of the NMDA receptor is minimally expressed in hippocampal synaptosomes, an effect that is accompanied by reduced expression of GluA1/GluA2 heteromers and an increase in GluA1/GluA1 homomers of the AMPA receptor [73]. These modifications in the ratio of glutamate receptors might contribute to the generation of the EPSP-LTP slope but prevent the proper functioning of NMDA receptors and the activation of second messengers. Furthermore, evidence has consistently shown that chronic epilepsy is associated with low levels of PKCγ in the DG [74,75]; this isotype is activated by Ca^2+^ and is required for LTP induction [76]. Moreover, a recent study described high levels of Fyn in the hippocampus of EPI rats [77]; although Fyn is commonly recognized as an inductor of LTP [78], the authors suggest that this enzyme may be able to impede hippocampal LTP when the HTR6/ERK1/2 pathway is activated [77]. Overall, the evidence suggests that the exacerbated glutamatergic neurotransmission in the DG of chronic EPI rats could not necessarily trigger robust LTP since several postsynaptic alterations in glutamate receptors and signaling pathways linked to LTP seem to be compromised in this pathology.

Regarding EPI animals treated with LEV, our data showed a decrease in EPSP-LTP. These results conflict with those reported by Sanchez et al. [34] and Ge et al. [18], who showed that LEV reverses the decrease in EPSP-LTP in the DG of an Alzheimer’s model and in the CA1 hippocampal region of EPI rats. The differences between these reports and our results may be due to differences in the animal model, disease progress, region registered or treatment scheme. However, our results are consistent with the hyperinhibition hypothesis (Figure 8). It has been reported that under pathophysiological conditions, LEV increased GABA release (Figure 8c) [36,79]. In addition, in EPI DG, the inhibitory interneurons, but not the principal cells, are primarily immunoreactive to the neuronal activity marker c-Fos [49]. Furthermore, SV2A protein was substantially increased and coexpressed with the GABA marker in the cell bodies and dendrites of hilar interneurons of mice with PTZ-induced seizures [54]. These three factors, the pathophysiology of neural tissue, neuronal activity and SV2A expression, might have a selective inhibitory effect that influenced the decrease in the EPSP-LTP slope of EPI animals treated with LEV (Figure 8c). Therefore, the high activity of GABAergic interneurons, augmented by LEV, could keep granular cells hyperpolarized and not allow the generation of EPSPs.

On the other hand, EPI rats treated with LEV exhibited a reduction in PS-LTP with respect to the nonepileptic animals, but it was not significant with respect to EPI rats. As we mentioned before, the postsynaptic alterations in glutamate receptors and signaling pathways could impede PS-LTP generation under EPI conditions, and therefore, LEV may not have had an effect. Otherwise, our PS-LTP data could explain previous reports with respect to the cognitive impairment presented in animal models and patients with TLE [80,81,82], as it has been postulated that the LTP corresponds to the cellular bases of memory processes. Although our results do not show positive effects on short-term plasticity and PS-LTP in rats treated with LEV, it cannot be overlooked that the drug may enhance cognitive processes in the long term, as was observed in a study conducted in children who were treated with LEV for benign epilepsy and showed an improvement in cognitive abilities [83].

## 5. Conclusions

Taken together, our results showed that TLE provoked profound changes in the basal excitability and synaptic plasticity of the DG and provide evidence that LEV has a long-lasting effect, reducing the basal excitability of granule cells in TLE rats. Although most alterations were not reestablished by LEV treatment, our study suggests that LEV may act as an effective antiseizure agent that potentiates inhibitory transmission, enhancing GABA release and suppressing the firing of glutamatergic neurons in the DG. This view is supported by EPSP-LTP data that reveal that LEV may have an effect on the hyperpolarization of granule cells in EPI rats. Our results do not support the role of LEV as a drug able to reestablish synaptic plasticity in TLE rats; nevertheless, further investigation is required to determine whether other factors (e.g., treatment duration or doses) could influence the effectiveness of LEV treatment. Moreover, epilepsy is a complex disorder that involves both neuronal inhibition and neuronal excitation; therefore, further studies are needed to better understand the complete mechanism of action of LEV.

## Figures and Tables

**Figure 1 brainsci-10-00634-f001:**
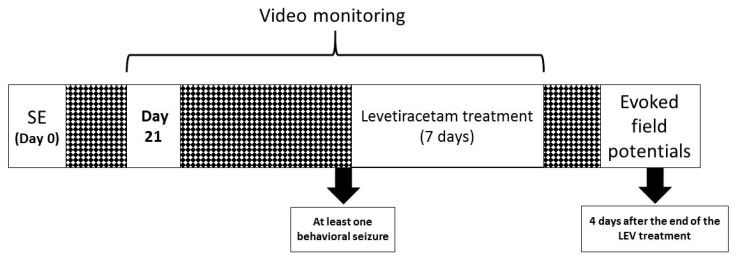
Experimental design. At time 0, status epilepticus (SE) was induced in male Wistar rats via lithium–pilocarpine administration. Three weeks after SE induction, rats were video monitored until the first spontaneous behavioral seizure was detected and then treated with levetiracetam (300 mg/kg/day) for one week. Four days after the end of the treatment, electrophysiological experiments were conducted; finally, animals were intracardially perfused, and brain dissection was performed.

**Figure 2 brainsci-10-00634-f002:**
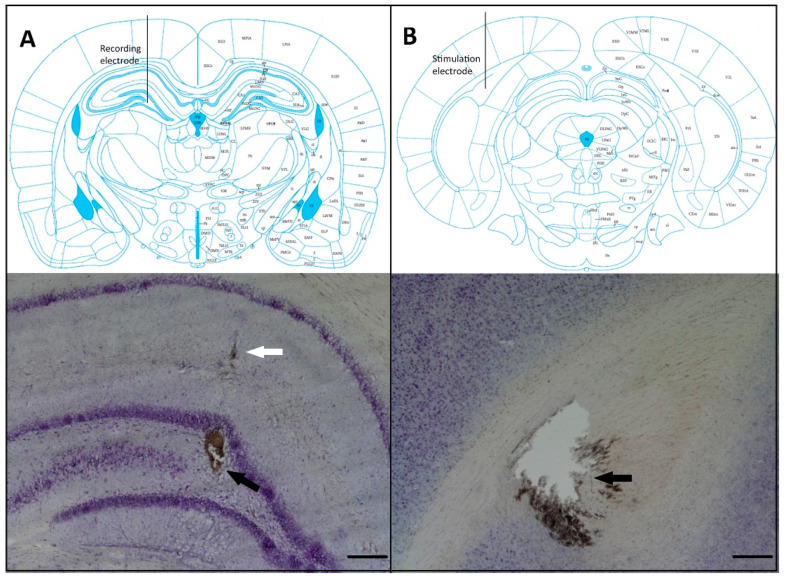
Recording and stimulation electrode placement. (**A**) The top panel shows a schematic drawing of coronal sections illustrating the recording electrode placement in the hilus of the dentate gyrus (from Paxinos and Watson [40]). The bottom panel shows a photomicrograph of a Nissl-stained section showing the recording site of the extracellular field potentials (black arrow); the white arrow indicates the trajectory of the electrode. (**B**) The top panel shows a schematic drawing of the coronal section illustrating the stimulation electrode placement in the perforant pathway. The bottom panel shows a photomicrograph of a Nissl-stained section depicting the stimulation site (black arrow). Scale bars 250 μm.

**Figure 3 brainsci-10-00634-f003:**
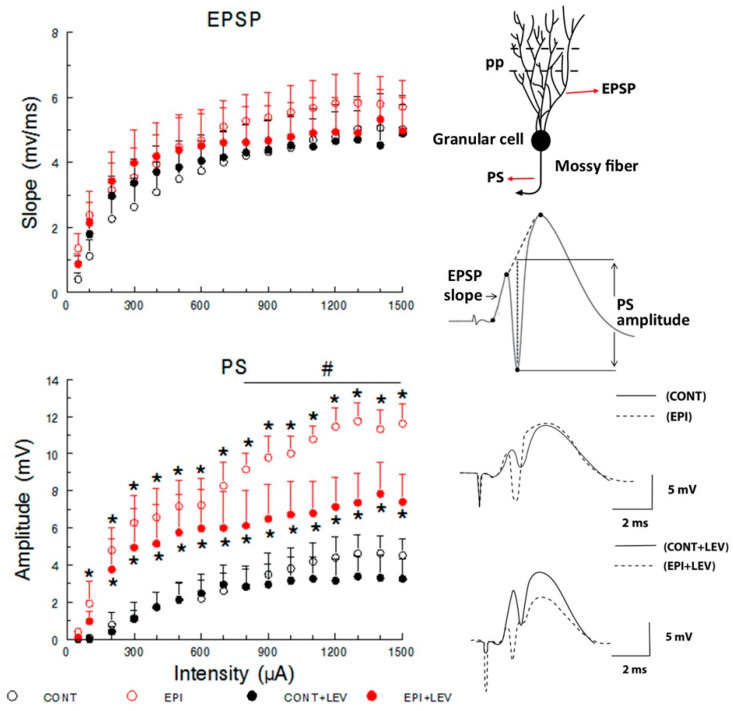
Input–output curve of the excitatory postsynaptic potential (EPSP) slope (**top**) and the population spike (PS) amplitude (**bottom**) in control (CONT) and epileptic (EPI) rats and control (CONT + LEV) and epileptic (EPI + LEV) rats after one week of levetiracetam (LEV) treatment (300 mg/kg/day). Field extracellular potentials were evoked in anesthetized rats by stimulation of the perforant path (pp) and were recorded in the dentate gyrus (**top right**). Data are presented as the means + standard error of the mean (S.E.M.), * *p* < 0.05 vs. CONT and CONT + LEV groups; # *p* < 0.05 vs. EPI + LEV group; three-way repeated-measures (RM) ANOVA followed by the Student–Newman–Keuls post-hoc test; *n* = 6 to 7 animals per group. At the (**top right**), a typical response to a high-intensity pulse, the slope of the rising positive phase (EPSP) and the negative superimposed PS are depicted. At the (**bottom right**), representative examples of traces showing the morphology of evoked field potentials are presented.

**Figure 4 brainsci-10-00634-f004:**
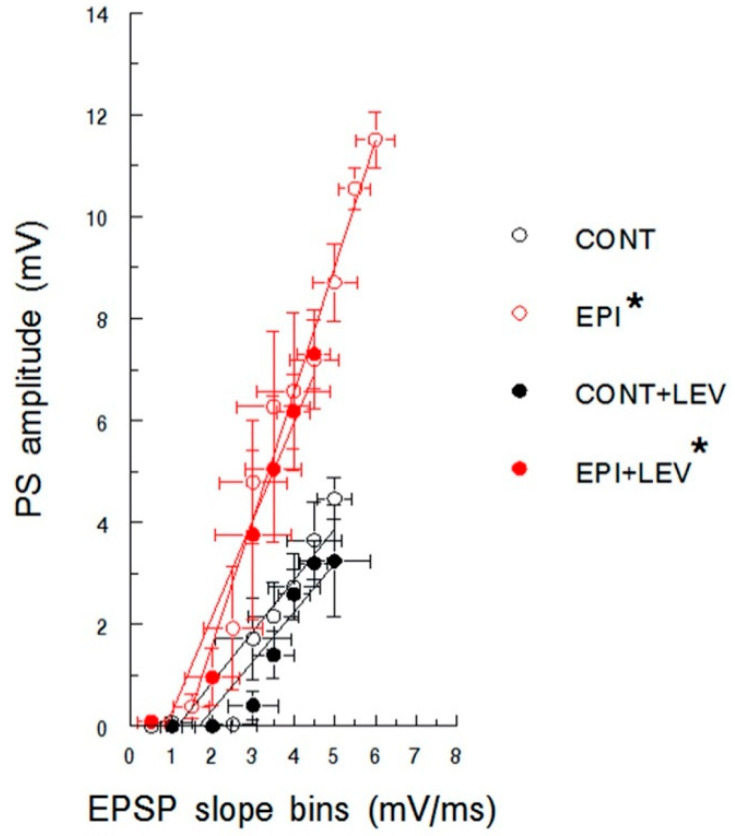
Excitability curve plotting population spike (PS) amplitude as a function of excitatory postsynaptic potential (EPSP) slope in control (CONT) and epileptic (EPI) rats and control (CONT + LEV) and epileptic (EPI + LEV) rats after one week of levetiracetam (LEV) treatment (300 mg/kg/day). Field extracellular potentials were evoked in anesthetized rats by stimulation of the perforant path and were recorded in the dentate gyrus. EPSP slopes were sorted into 0.5 mV/ms bins. Data are presented as the means ± S.E.M. * *p* < 0.05 vs. CONT and CONT + LEV groups, Student’s *t*-test for comparing slopes.

**Figure 5 brainsci-10-00634-f005:**
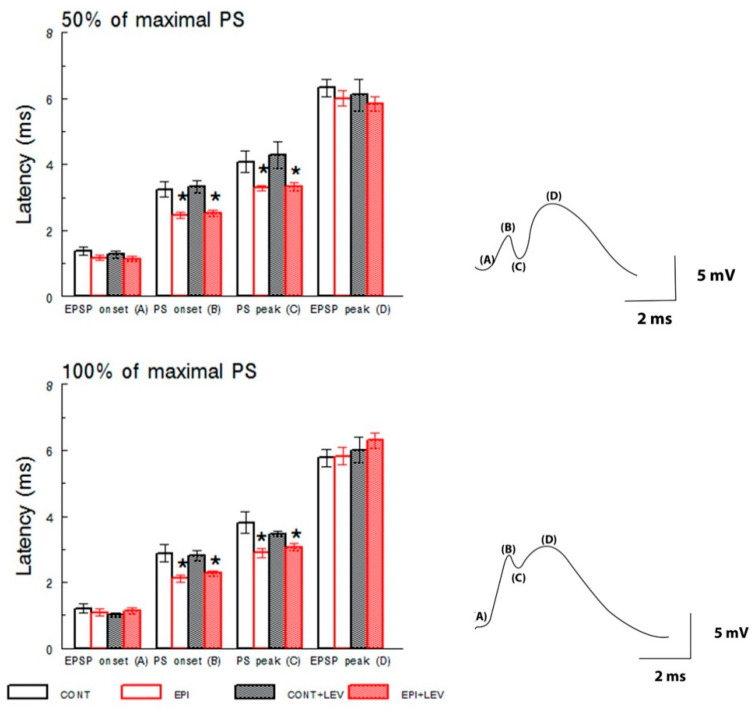
Latencies of excitatory postsynaptic potentials (EPSPs) and population spike (PS) at intensities that produced 50% (**top**) or 100% (**bottom**) of the maximal PS according to the input–output curve in control (CONT) and epileptic (EPI) rats and control (CONT + LEV) and epileptic (EPI + LEV) rats after one week of levetiracetam (LEV) treatment (300 mg/kg/day). Field extracellular potentials were evoked in anesthetized rats by stimulation of the perforant path and were recorded in the dentate gyrus. Data are represented as the means ± S.E.M., * *p* < 0.05 vs. CONT and CONT + LEV groups; two-way ANOVA followed by the Student–Newman–Keuls post-hoc test; *n* = 6 to 9 animals per group. On the right, representative examples of traces showing the latencies of evoked field potentials are presented. EPSP onset (A), PS onset (B), PS peak (C) and EPSP peak (D).

**Figure 6 brainsci-10-00634-f006:**
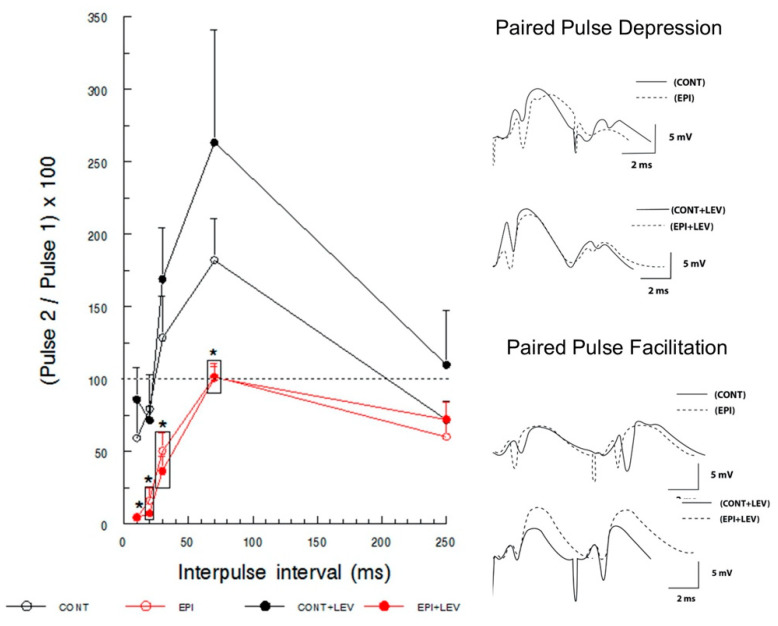
Paired-pulse depression and facilitation of population spike (PS) in control (CONT) and epileptic (EPI) rats and control (CONT + LEV) and epileptic (EPI + LEV) rats after one week of levetiracetam (LEV) treatment (300 mg/kg/day). Field extracellular postsynaptic potentials were evoked by perforant path stimulation and recorded in the dentate gyrus at intensities that produced 100% of the maximal PS from the input–output curve with interpulse intervals (IPIs) of 10, 20, 30, 70 and 250 ms. Data are represented as the means + S.E.M. of paired-pulse percentages ((pulse 2 amplitude/pulse 1 amplitude) × 100); a percentage < 100% reflects depression and a percentage > 100% reflects facilitation. * *p* < 0.05 vs. CONT and CONT + LEV groups. Three-way RM ANOVA followed by the Student–Newman–Keuls post-hoc test; *n* = 7 to 9 animals per group. On the right, representative examples of traces showing the depression and facilitation of evoked field potentials are presented.

**Figure 7 brainsci-10-00634-f007:**
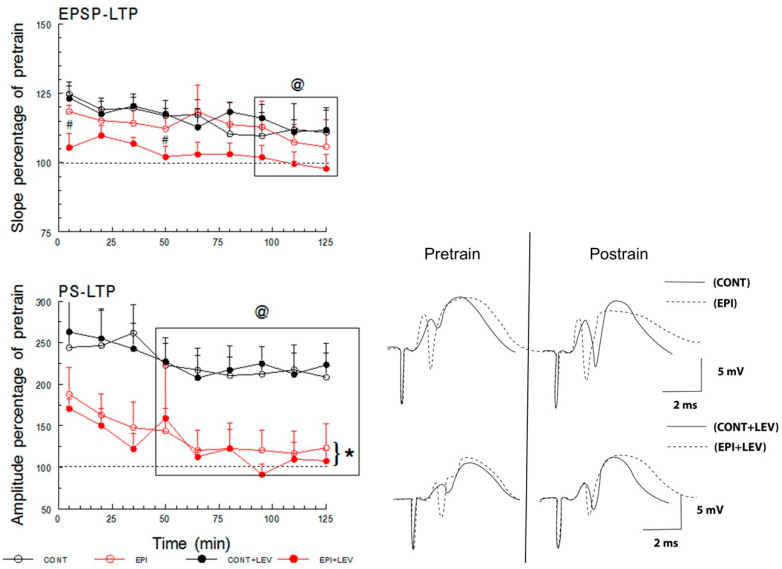
Long-term potentiation (LTP) of the excitatory postsynaptic potential (EPSP) and population spike (PS) recorded in the dentate gyrus. LTP of the EPSP slope (**top**) and PS amplitude (**bottom**) in control (CONT) and epileptic (EPI) rats and control (CONT + LEV) and epileptic (EPI + LEV) rats after one week of levetiracetam (LEV) treatment (300 mg/kg/day) are presented as percentages of baseline (pretrain) values. LTP was induced with trains of 1500 µA, and pre- and post-train field extracellular potentials were evoked at intensities that produced 50% of the maximal PS from the input–output curve. Data are represented as the means + S.E.M., # *p* < 0.05 vs. EPI group; @ *p* < 0.05 vs. time = 5 min; * *p* < 0.05 vs. CONT and CONT+LEV groups. Three-way RM ANOVA followed by the Student–Newman–Keuls post-hoc test; *n* = 6 to 7 animals per group. Representative examples of traces showing pre- and post-train field extracellular potentials are presented in the bottom right.

**Figure 8 brainsci-10-00634-f008:**
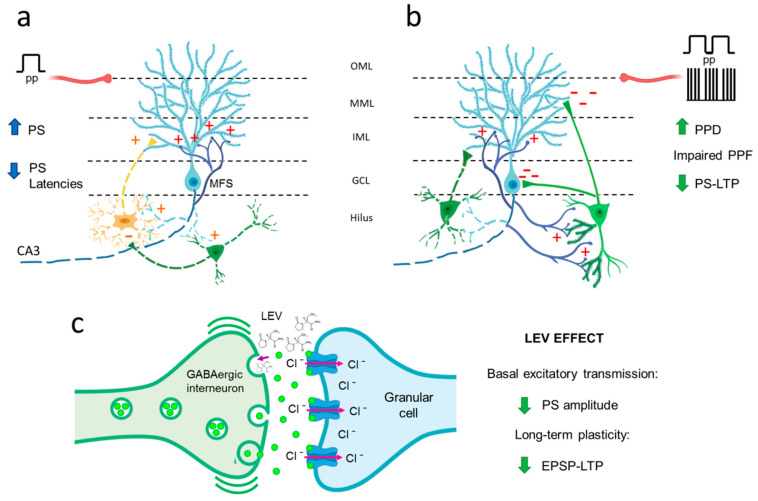
Hypothetical schematic representation of the dentate gyrus (DG) environment in TLE rats and the levetiracetam (LEV) mechanism of action. (**a**) As a consequence of the death of mossy cells (yellow dotted), GABAergic cells (green dotted), and pyramidal CA3 cells (not shown), the granule cells (blue cell) remains without postsynaptic targets; their axons then “sprout” and innervate the dendrites of other granule cells (not shown) and themselves, generating the mossy fiber sprouting (MFS) and inducing granule cell hyperexcitability [1,43]. The clear augment of the population spike (PS) amplitude and the significant reduction in the onset- and peak-PS latencies in the basal excitatory transmission of TLE rats supports this phenomenon. (**b**) The hyperexcitability of granule cells in TLE could over-activate the surviving GABAergic interneurons (right green), increasing local inhibition. In addition, the loss of interneurons (left green dotted) provokes aberrant connections between the axons of granule cells and remaining GABAergic interneurons, which, in turn, may develop new dendritic spines and, thus, new synaptic contacts [48,51,52]. All this together would promote a hyperinhibitory environment; this could explain the changes in the short and long-term synaptic plasticity in TLE rats, such as strong depression (PPD) and the absence of facilitation (PPF) by paired pulses, and the decrease in PS long-term potentiation (PS-LTP). (**c**) The partial recovery of PS amplitude in basal excitatory transmission and the decrease in excitatory postsynaptic potential (EPSP)-LTP slope by LEV, could be associated with the potentiation of the GABAergic signaling through the increase in the release of γ-aminobutyric acid (GABA), suggesting that LEV may act as an effective antiseizure agent that suppresses the firing of glutamatergic neurons in the DG. OML: outer molecular layer; MML: middle molecular layer; IML: inner molecular layer; GCL: granule cell layer; pp: perforant pathway (red axon).

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
