# Peer review of "Levetiracetam Reduced the Basal Excitability of the Dentate Gyrus without Restoring Impaired Synaptic Plasticity in Rats with Temporal Lobe Epilepsy"

_brainsci, 2020, doi:10.3390/brainsci10090634_

Round 1

Reviewer 1 Report

In this manuscript, the authors conducted field potential recordings from the dentate gyrus of hippocampus to investigate the excitability, short-term and long-term synaptic plasticity and the effects of Levetiracetam (LEV) on those parameters in the pilocarpine model of temporal lobe epilepsy (TLE). Their results showed that 1) TLE increased excitability (indicated by increased PS amplitude) and synaptic depression, and decreased paired-pulse facilitation and LTP; 2) LEV treatment decreased the excitability, but did not have significant long-lasting effects on synaptic plasticity.

Overall, the manuscript was very well written with sound experiment design and convincing results. I recommend it for publication.

Author Response

We greatly appreciate receiving positive comments about our manuscript from the Reviewer. 

Reviewer 2 Report

Dear Editor

The study by González-Hernández et al confirmed that TLE provoked profound changes in the basal

excitability and synaptic plasticity of the dentate gyrus (DG) and provides an evidence that levetiracetam has a long-lasting effect, reducing the basal excitability of granule cells in TLE rats. Results from this study suggests that levetiracetam may act as an effective antiseizure agent that potentiates inhibitory transmission, enhancing GABA release and suppressing the firing of glutamatergic neurons in the DG.

The design of the study and the technical quality of the work are convincing and results are of general interest. The manuscript is well-written and easy to follow. Authors used correct statistical approaches in analyzing the results and the data is well-presented. The conclusion is supported by the data and the discussion was through unbiased comparisons with up-to-date and relevant literature. I personally like Fig 8 since it provides a schematic representation of the DG environment in TLE rats and the levetiracetam mechanism of action. This will be of great interest for Brain Sciences readers and researchers within the field of epilepsy in general.

I would recommend this manuscript for publication once the author addressed the below mentioned minor point regarding the introduction.

Minor:

Starting with a more general introduction about TLE and cognitive decline, TLE prevalence and epidemiology, the need to identify new drug targets due to the increased resistance which reached about 30% of the cases and so, will the help the introduction to flow better. These are some of the suggested references:

https://www.ncbi.nlm.nih.gov/pmc/articles/PMC5898178/

https://www.ncbi.nlm.nih.gov/pubmed/28715131

https://www.ncbi.nlm.nih.gov/pubmed/29288503

Best

Author Response

We greatly appreciate receiving positive comments about our manuscript from you. We took into careful each comment and the manuscript has now been revised. Seven new references were added to make the appropriate corrections in the introduction section. In the introductory section, the first paragraph (lines 35 to 43) and part of the second (lines 44 to 46) were modified to include a more general introduction on the epidemiology of TLE, the cognitive impairment in TLE, and the need to investigate new drug targets for treatment of epilepsy and its comorbidities, since epileptic seizures are not controlled with medical treatment in up to 30% of patients.

We hope that you will find the manuscript improved and acceptable for publication in Brain Sciences.

Thank you